# Digital Tools for Revealing and Reducing Carbon Footprint in Infrastructure, Building, and City Scopes

**Jiayi Yan** [1] , **Qiuchen Lu** [1,*], **Junqing Tang** [2], **Long Chen** [3], **Jingke Hong** [4] **and Tim Broyd** [1]

1   The Bartlett School of Sustainable Construction, University College London, Gower Street, London WC1E 6BT, UK; j.yan.21@ucl.ac.uk (J.Y.); tim.broyd@ucl.ac.uk (T.B.)
2   School of Urban Planning and Design, Shenzhen Graduate School, Peking University, Shenzhen 518055, China; junqingtang@pku.edu.cn
3   School of Architecture Building and Civil Engineering, Loughborough University, Loughborough LE11 3TU, UK; l.chen3@lboro.ac.uk
4   School of Urban Planning and Design, Chongqing University, Chongqing 400044, China; hongjingke@cqu.edu.cn
*   Correspondence: qiuchen.lu@ucl.ac.uk

**Abstract:** The climate change issue has been striking and bringing pressure on all countries and industries. The responsibility of the Architecture, Engineering, Construction and Facility Management (AEC/FM) industry is heavy because it accounts for over one-third of global energy use and greenhouse gas emissions. At the same time, the development of digital technology brings the opportunity to mitigate environmental issues. Therefore, this study intended to examine the state-of-the-art of digital development and transformation in the AEC/FM industry by collecting and reviewing the developed digital carbon footprint analysis tools in infrastructure, building, and city scopes. Specifically, this study (1) generated a review methodology for carbon footprint analysis results; (2) demonstrated the review results from the infrastructure, building, and city scopes, analysed and compared the results crossing the scopes from four aspects: carbon footprint analysis strategy, standards and protocols, rating systems, and general development level of digital tools; and (3) discussed the potential directions in the industry to address the environmental issues. This study indicated that the digitalisation level regarding carbon-related areas is still at an early stage, and efforts should be taken both academically and practically to drive the digital development confronting the harsh climate change issue.

**Keywords:** carbon footprint (CF); greenhouse gas (GHG) emissions; carbon management; carbon accounting; digital transformation; digitalisation; infrastructure; BIM

## 1. Introduction

The consequences of deteriorated climate change are threatening all countries. Each nation is taking urgent steps to address environmental issues. For example, the United Kingdom (UK), the United States (US), and Japan have set up the goal to reduce greenhouse gas (GHG) emissions and reach the carbon net zero status by 2050 [1–3]. The urgent climate emergency pressed by the 2021 United Nations Climate Change Conference (COP26) has required all industries to step further beyond the current set of policy measures and has demanded all industries to provide even "smarter" and "clear" metrics that would be equitable to meet the Earth's ecological limits [4].

The construction industry has always been one of the largest emitters of carbon dioxide. Research conducted by [5] suggested that the Western infrastructure stock using the existing technologies could cause about 350 Gt $CO_2$ only from the construction material production, which corresponds to 35–60% of the remaining carbon budget given the 2 °C limit until the year 2050. In the United States, an early report showed that the construction industry was responsible for 131 million metric tons (MT) of $CO_2$, which ranked third place in the carbon

dioxide emitter in the nation [6]. In its long service lifetime, infrastructure is critical for continuously satisfying human needs for water, energy, transportation, and communication. Therefore, the decision made for infrastructure highly likely has a huge impact on the carbon dioxide emissions of a country or society over a long time [7]. Buildings, and the cities consisting of them, are probably the most human-interacted premises, contributing one-third of all GHG emissions globally [8,9]. However, the building and city sectors were also suggested to be the most potential areas to mitigate carbon emissions [9]. According to the research conducted by [10], the capital carbon consumption for a typical building (e.g., a residential building) ranged from 0.3–3.0 $tCO_2$ e/$m^2$, while a typical infrastructure project (i.e., a highway bridge) ranged from 2.9–63.6 $tCO_2$ per linear meter.

To calculate and reduce the carbon footprint (CF) emissions in the AEC/FM sector, assessing methods, such as the life cycle assessment (LCA) for manufacturing products [11] and thus the whole-building LCA (WBLCA), [12] and infrastructure-oriented carbon management methods such as PAS 2080 by [13] came out gradually. Discussions and comparisons of the methods have been conducted in specified scopes such as infrastructure, building, and city [14–18]. In these previous studies, the importance and expectation to use efficient and mature commercialised digital tools, platforms, or systems have been all emphasised by researchers to help with the automation and decision making for carbon management issues [14,16,19]. Moreover, the tide of industry 4.0 and the introduction of more and more cyber concepts such as building information modelling (BIM), digital twin, Internet of Things (IoT), and artificial intelligence (AI) have come. It has been suggested that these technologies can also be utilised to tackle carbon emission issues, which was testified in other relatively advanced disciplines such as chemical engineering [20]. However, currently in the AEC/FM sector, research questions such as "where are we at regarding the digitalisation of carbon emission assessment?" and "what are the future trends towards digital tools development and how could we apply technology on this subject?" remain unsolved.

To address the research gaps, this paper presents a complete review of the digital tools in both academia and the industry that provide CF calculations in the AEC/FM sector, especially targeting infrastructure, building, and city scopes. The following sections of this paper demonstrate the findings. Section 2 introduces the methodology of this review study. Section 3 illustrates the review results from the infrastructure, building, and city scopes. In addition, a cross-scope analysis in terms of CF analysis strategy, standards and protocols, rating systems, and development level synthesises the tools in three scopes. Lastly, Section 4 discusses the potential development trends and challenges for the tools in the AEC/FM industry.

## 2. Methodology

In order to perform a comprehensive review of the CF-calculation-related digital tools and approaches in the AEC/FM sector, this study developed the methodology with four steps included (Figure 1).

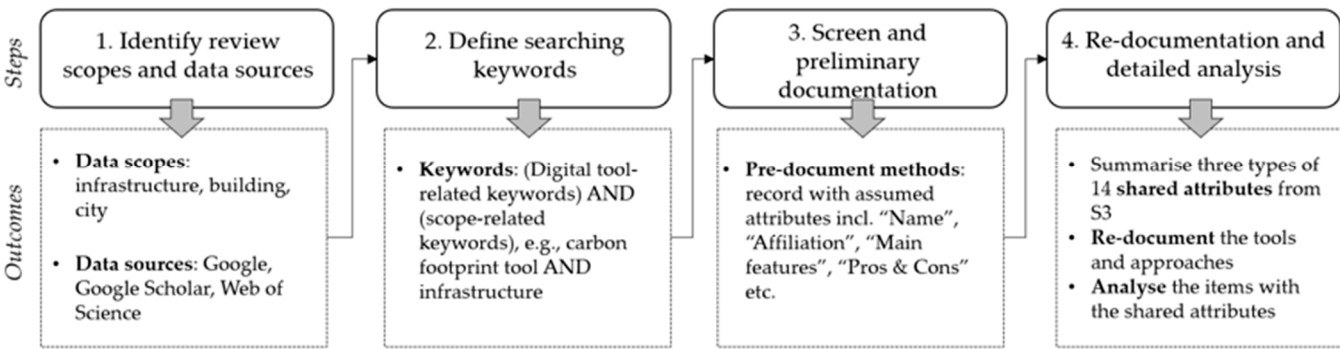

**Figure 1.** The four-step review methodology.

The first step was to specify the review data scopes and sources. There were two considerations. Firstly, given the features of the assets including but not limited to the complexity of building structures and elements and operation synergies of the built environment systems [16], this review targeted three scopes: infrastructure, building, and city. Secondly, to thoroughly examine the status quo of digital tools comprehensively, publicly available tools including both commercial tools and products and academic tools needed to be involved. Main commercial tools should especially not be overlooked as they may tend to be efficient and mature [16,21]. Considering the review scopes, Google search engine was considered the primary data source and Web of Science and Google Scholar were used for supplementary searching.

The second step was to define the searching keywords. Two types of keywords were connected with the "AND" operator: digital-tool-related words and scope-related words. For instance, selected keywords include "carbon footprint tool AND infrastructure", "carbon footprint tool AND buildings", and "carbon footprint tool AND city".

The third step was to screen the search results and preliminarily document the selected tools. During the screening process, if there was neither a description regarding the full or partial preliminary documenting attributes such as "Name, "Affiliation", "Main features", and "Pros and Cons" nor a demo available, the searched tools were excluded from the documentation. After this step, a spreadsheet with the preliminary documenting attributes of all selected digital tools within the three scopes (i.e., infrastructure, building, and city) was created.

Based on the outcome from step 3, the fourth step was to re-document the selected tools and analyse them in detail. In the first place, three types of attributes were generated as follows:

- Basic information: it describes the information of each tool's developing origin and status.
- CF analytical information: this attributes group indicates how each digital tool measures the CF's environmental impact and the scopes of covered emissions. Specifically, the inclusions of embodied emissions and operational emissions were examined. In this study, the embodied emissions refer to the CF generated before the completion of the construction [14], such as the carbon emissions of producing the construction materials and the emissions of transporting the materials from the factories to construction sites by workers. Operational emissions refer to the CF generated during the operation and maintenance phase of the assets, such as carbon emissions to heat and ventilate the buildings, maintenance activities by workers, and end-of-life disposal of the buildings [14].
- Digitalisation information: this attributes group is about how the tool has been connected to the development of updated technology applications. One of the most important considerations for this attributes group was whether the tool has been potentially designed to adopt building information modelling (BIM), a widely accepted digital approach in the AEC/FM industry [16].

Specifically, there are 14 descriptive attributes belonging to the abovementioned three types in Table 1 [21]. Then, each tool was documented with the 14 attributes. Finally, two summarised spreadsheets of the infrastructure and building scopes were developed, respectively. The results of the city scope are described without a spreadsheet because the tools are very limited aiming for this scope. Then, the results were analysed and compared by crossing the three scopes to provide synthesised conclusions in terms of CF analysis strategy, standards and protocols, rating systems, and the development level of the digital tools in general.

**Table 1.** Summary of the review attributes.

| Type | Attributes | Contents |
|---|---|---|
| Type 1: Basic Information | Name | The name of the developed digital tool for CF calculation |
| | Affiliation | The organisations or companies that develop and ooperatethe tool |
| | Development Region | The region from which the tool originates |
| | Specified Industry | More specified fields that the tool targets |
| | Operation Type | Whether the tool is commercial (charged)/non-profit (free)/academic (developed and free for academic purposes) |
| | Current Availability | Whether the tool is still available regardless of the operation type |
| Type 2: CF Analytical Information | Standard Coverage | The CF calculation standards that the tool complies |
| | Analysis Strategy | The CF calculation methods from the life cycle analysis perspectives |
| | Embodied Emissions Included | Whether the analysis strategy includes embodied emissions |
| | Operational Emissions Included | Whether the analysis strategy includes operational emissions |
| Type 3: Digitalisation Information | BIM adaptation | Whether the tool "talks" to the BIM-related software (e.g., calculate using BIM files or be the add-in in the BIM software) |
| | Release Form | Whether the tool is released in spreadsheet-based/web-based (cloud-based)/standalone/add-in form |
| | Result Presentation | How the calculation results are presented (e.g., simplified numbers or reports with diagrams) |
| | Digital Environment | Whether the tool provides the 3D digital environment of the calculated built environment |

## 3. Results and Analysis

*3.1. Review Results in Infrastructure, Building, and City Scopes, Respectively*

3.1.1. CF Calculation Tools in the Infrastructure Scope

In this scope, there are six tools that have been developed targeting the infrastructure CF calculation as shown in Table 2. In general, the choices of digital tools are limited considering different application scenarios.

From the basic information (Type 1 attributes) perspective, it can be observed that there are four digital tools that are currently available to the public aiming at different specific infrastructure types. For example, the Rail Carbon Tool (RCT) was developed by Rail Safety and Standards Board (RSSB) in the UK, which focuses on the whole-life CF calculation for railways [22]. The Highways Agency Carbon Calculator for Construction developed by the UK National Highway and asPECT developed by TRL focus on highway infrastructure CF. Particularly, asPECT calculates life cycle GHG emissions in asphalt used for highways [23,24]. Infrastructure LCA from One Click LCA is the only digital platform that was designed and developed for a variety of infrastructures including airports, bridges, transmission system flood alleviation schemes, park ride schemes, etc. [25], which covers relatively comprehensive application scenarios.

**Table 2.** Summary of digital tools in infrastructure scope.

| Name | Basic Information | | | | | | CF Analytical Information | | | | Digitalisation Information | | | Ref. |
|---|---|---|---|---|---|---|---|---|---|---|---|---|---|---|
| | Affiliation | Region | SpecifiedIndustry | Operation Type | Current Availability | Standard Coverage | Analysis Strategy | Embodied Emissions | Operational Emissions | BIM Adaptation | Release Form | Result Presentation Form | Digital Environment | |
| Rail Carbon Tool (RCT) | RSSB (Powered with Atkins) | UK | Railway | Commercial/non-profit | Yes | PAS 2080 GHG Protocol Scope 1, 2, 3 | Whole-life Carbon | Yes | Yes | N/A | Web-based | Summarised report with 2D diagrams | Not available | [22] |
| The Highways Agency Carbon Calculator for Construction | National Highway | UK | Highway | Commercial/non-profit | Yes | PAS 2050 GHG Protocol Scope 1, 2, 3 | Construction phase analysis | Yes | No | No | Spreadsheet-based | Summarised report with 2D diagrams | Not available | [26] |
| asPECT | TRL | UK | Asphalt used on highways | Commercial/non-profit | Yes | PAS 2050 | Construction phase analysis | Yes | No | No | Standalone | Summarised Report in numbers | Not available | [23] |
| Infrastructure LCA | One Click LCA | US | Infrastructure in general | Commercial | Yes | EN 17472:2021, PAS 2080 | LCA | Yes | Yes | Yes | Web-based/Add-in | Summarised report with 2D diagrams | Yes | [25] |
| Carbon calculator design tool for bridges | British Constructional Steelwork Association Ltd. (BCSA), Tata Steel and Atkin | UK | Steel-concrete composite typical bridge | Academic | Not available | ISO 14040 | LCA | Yes | Yes | No | Standalone | Summarised report with 2D diagrams | Not available | [27] |
| Carbon Footprint Estimation Tool (CFET) | Environmental Inc.; Unv. Of Maryland | US/Canada | Railway | Academic | No | IPCC Guidelines GHG reduction policies | Construction phase analysis | Yes | No | N/A | Standalone | Summarised results in numbers | Not available | [28] |

From the CF analytical information (Type 2 attributes) perspective, the reviewed tools can be categorised using two analysis methods. One of the methods is to calculate the embodied carbon emissions related to all the materials and activities that happen during the infrastructure's construction phase, which is also regarded as the emissions measured during the "cradle-to-completed construction" stage [14]. For example, the highway calculator from National Highway identifies the CF calculation scopes including the emissions in raw materials (e.g., concrete, cement and binders, and reinforcement steel), emissions from construction sites or maintenance areas (e.g., use of fuels, energy, or water, business, and employee transport), and emissions from the processing of waste at disposal facilities [26]. The calculator asPECT emphasises the life cycle CF of producing asphalt used for highway construction specifically [24]. The other method is to conduct the whole-life CF calculation (i.e., LCA) including both embodied and operational emissions, which cover the emissions measured during the "cradle-to-grave" stage [14] like RCT by [22], Infrastructure LCA by [25], and Carbon Calculator design tool for bridges by [27].

From the digitalisation information perspective (Type 3 attributes), the development is limited in the infrastructure scope generally. First, most of the digital tools are not adapted to BIM, which is the foundation for digitalisation in AEC/FM industry [16]. Additionally, the results are presented in the form of plain breakdown numbers [23,28] or charts [22,26,27]. No matter whether the information provided by the tools is sufficient for the user, these two issues can potentially become the barrier to take the carbon emission issue into account from the design to operation stages by stakeholders. In addition, researchers indicated that the support of digital environments (e.g., the adoption of BIM model) can be helpful to define the assets' structure and the system boundaries [16], whereas most of the digital tools are not capable of this in the infrastructure scope. Infrastructure LCA is the only tool that is both able to integrate with BIM and provide a digital environment for users to refer to. As for the release form, it can be observed that the tools which have been developed earlier were released as standalone software [23,27,28], while the latter tools tend to be in web-based form, add-in form, and spreadsheet form [22,25,26].

Moreover, the CF issue of infrastructure has long been a target academically. Other than the scopes of railways, highways, and bridges that are discussed in Table 1, a wider range of infrastructure scopes' CF calculation methods have been addressed, such as airport, port, underground utility, water supply infrastructure, etc. [29–32]. More deliberated CF calculation methodologies have been studied. For example, Ref. [33] used a hybrid of LCC analysis of the LCA method to reduce carbon emissions from maintenance and rehabilitation of highway pavement. In addition, the relatively lagged development of digitalisation in the UK's rail CF calculation tools was already identified by [19]. However, these outcomes have not been transformed into mature digital and applicable tools that can be adopted by practitioners widely.

### 3.1.2. CF Calculation Tools in the Building Scope

In the building scope, there are 18 digital tools or platforms that were reviewed (Table 3). Compared to the infrastructure scope, there are more choices targeting different application scenarios.

From the basic information (Type 1 attributes) perspective, the tools and platforms have been developed targeting multiple categories of building assets, including both residential and non-residential buildings. Some of them can also be used for infrastructure CF calculation [34,35], but they are not designed deliberately for infrastructures. Moreover, tools for building assets that are constructed with certain materials such as wood and stone have been especially developed [36,37].

**Table 3.** Summary of digital tools in building scope.

| Name | Basic Information | | | | | | CF Analytical Information | | | | Digitalisation Information | | | Ref. |
|---|---|---|---|---|---|---|---|---|---|---|---|---|---|---|
| | Affiliation | Region | Specified Industry | Operation Type | Current Availability | Standard Coverage | Analysis Strategy | Embodied Emissions | Operational Emissions | BIM Adaptation | Release Form | Result Presentation Form | Digital Environment | |
| Athena Impact Estimator for Buildings | ATHENA Sustainable Material Institute | North America | Building | Commercial/ non-profit | Yes | ISO 14040 and 14044 series | WBLCA | Yes | Yes | No | Standalone | Summarised report with 2D diagrams | No | [34] |
| Athena EcoCalculator for Assemblies | ATHENA Sustainable Material Institute | North America | Building | Commercial/ non-profit | Yes (but no longer maintained) | ISO 14040 and 14044 series | LCA | Yes | Yes | No | Spreadsheet-based | Not Available | No | [34] |
| eToolLCD | eTool | UK | Building, in-frastructure | Commercial | Yes | EN 15978 and ISO 14044 | LCA | Yes | Yes | Yes | Web-based | Summarised report with 2D diagrams | Yes | [35] |
| Carbon Calculator | Forest Pennant (Natural Stone Specialist) | UK | Building (stonework) | Commercial/ non-profit | No | PAS 2050 | Construction phase analysis | Yes | No | No | Spreadsheet-based | Not Available | No | [37] |
| Embodied Carbon and Energy Efficiency Tool | Thornton Tomasetti | UK | Building | Academic | No (2014) | Inventory of Carbon & Energy (ICE) | N/A | Yes | No | N/A | Rhino | Visualised in 3D, (parameter design) | Yes | [38] |
| OERCO2 | Erasmus+ | EU | Building | Academic | Yes | IPCC 100a methodology | LCA | Yes | No | Not now | Web-based | Summarised Report in numbers | No | [39] |
| WoodWorks Carbon Calculators | WoodWork | US | Wood building | Commercial | Yes | National Design Specification® (NDS®) for Wood Construction | Construction phase analysis | Yes | No | No (only wood elements in .rvt) | Not Available | Not Available | Not Available | [40] |
| One-Click LCA | One Click LCA | Global | Building | Commercial | Yes | EN 15978, EN 15804, EN 15942, ISO 21931-1, ISO 21929-1, ISO 21930, | WBLCA | Yes | Yes | Yes | Add-in/Standalone | Summarised report with 2D diagrams | Yes | [41] |
| Tally | Tally (stewarded by Building Transparency) | US | Building | Commercial | Yes | EN 15643, EN 15978, ISO 14040 and 14044 | WBLCA | Yes | Yes | Yes | Add-in | Summarised report with 2D diagrams | Yes | [42] |
| Embodied Carbon in Construction Calculator (EC3) | Building Trans-parency | US | Building materials | Commercial/non-profit | Yes | Sorting and visualization of EPDs | A compre-hensive product database | Yes | No | Has API | Web-based | Summarised report with 2D diagrams | No | [43] |

**Table 3.** *Cont.*

| Name | Basic Information | | | | | CF Analytical Information | | | | | Digitalisation Information | | | Ref. |
|---|---|---|---|---|---|---|---|---|---|---|---|---|---|---|
| | Affiliation | Region | Specified Industry | Operation Type | Current Availability | Standard Coverage | Analysis Strategy | Embodied Emissions | Operational Emissions | BIM Adaptation | Release Form | Result Presentation Form | Digital Environment | |
| IMPACT | BRE | UK | Building | Commercial | Yes | EN 15804 | LCA | Yes | No | Yes | Web-based/ Add-in | Summarised report with 2D diagrams | Yes | [44] |
| e2CO2Cero | Basque Government | Spain | Building | Commercial | Yes | ISO14040: 2006. ISO14044: 2006. ISO 14025: 2006 | LCA | Yes | Yes | No | Web-based | Summarised report with 2D diagrams | No | [45] |
| The Structural Carbon Tool | The Institute of Structural Engineers | UK | Building | Commercial/non-profit | Yes | BS EN 15978, BS EN 15804 | LCA | Yes | No | No | Spreadsheet-based | Summarised report with 2D diagrams | No | [46] |
| Build Carbon Neutral | University of Texas at Austin, University of Washington | US | Building | Academic | Yes | Inventory of Carbon & Energy (ICE) | Construction phase analysis | Yes | No | No | Web-based | Summarised Report in numbers | No | [47] |
| a BIM Tool | National Cheng Kung University, Taoyuan Innovation Institute of Technology | China (Taiwan) | Building | Academic | No | BIM-BCF (building carbon footprint) evaluation | Building life cycle | Yes | Yes | Yes | Add-in | Not Available | Yes | [48] |
| BuildingScope™ | Clean Metrics 2.0 | US | Building | Commercial | No (2011) | ISO 14040 series, PAS 2050, GHG Protocol | LCA | Yes | No | No | Web-based | Summarised report with 2D diagrams | No | [49] |
| CFCCP | American University of Beirut, Lebanon | Lebanon | Building | Academic | No (2011) | Renewable Energy Laboratory (NREL) | Construction phase analysis | Yes | No | No | Standalone | Summarised report with 2D diagrams | No | [50] |
| Environment Agency Carbon Calculator | Energy Agency | UK | Building | Not Available | Not Available | Not Available | Construction phase analysis | Yes | No | No | Spreadsheet-based | Not Available | No | [51] |

From the CF analytical information (Type 2 attributes) perspective, the CF calculation methods can still be categorised into "cradle-to-completed construction" measurement and "cradle-to-grave" measurement depending on the standards the tools comply with. However, on top of the LCA method (i.e., cradle-to-grave measurement), whole-building life cycle assessment (WBLCA) has been iterated to address the CF issue in the building sector. WBLCA is better to monitor the carbon emissions from the product and construction stage, operational stage, maintenance stage, and disposal stage [16]. Dominant digital tools such as Athena Impact Estimator for Buildings, One Click LCA, and Tally have adopted WBLCA as the calculation methodology.

From the digitalisation information perspective (Type 3 attributes), the development is more advanced than the infrastructure scope. For example, multiple tools were designed and developed to integrate with BIM software such as One Click LCA, Tally, eToolLCD, and IMPACT. Among them, Tally was developed directly for the convenience of Revit. The feature of BIM adaptation not only guarantees the availability of a digital environment for accurate analysis but also facilitates the data interoperability through the building assets' life cycle by sharing the environmental impact data [19]. In addition, researchers developed a novel demo that brought the embodied carbon emission database into a format that the architectural design tool Grasshopper can recognise so that the embodied carbon emissions results can be visualised in a 3D model of different building structures driven by design parameters [38]. Although the demo was at the pilot stage, the visualisation feature can be an empirical reference.

Furthermore, there are several digital tools that have noticeable characteristics. For example, OERCO2 is a project outcome supported by the EU. The tool was designed for non-specialised users to estimate residential buildings' CF [39]. Embodied Carbon in Construction Calculator (EC3) provides an open-source and large Environmental Product Declarations (EPDs) database for users to measure embodied emissions based on building material quantities from construction estimates and/or BIM models. EC3 was released in the form of an application programming interface (API) that can be integrated with many software products [40]. These characteristics are unique targeting different groups of industry users. In general, while there are still challenges such as calculation methods, building structure complexity, digital and technological capability, etc. [16,18,19] that remain to be solved, users can have various choices for different use scenarios.

### 3.1.3. CF Calculation Tools in the City Scope

City (or urban) CF is regarded as a thorough assessment of GHG emissions from an urban system [52]. The calculation results of city CF would drive and affect decision making and policy making greatly [52]. Therefore, the city-level carbon issue has long been critical both academically and practically. Steps have been taken to tackle the issue. For example, British Standards Institute (BSI) and World Resources Institute (WRI) published PAS 2070 and Global Protocol for Community scale GHG emissions (GPC) in the year of 2014, respectively, to guide the city-level CF calculation [53,54]. Based on the standards, the carbon emissions of cities such as London, Madrid, and Beijing have been assessed [55,56]. Moreover, researchers have generated iterated methods. For instance, Ref. [57] developed the concept of city carbon map, and [17] concluded the three mainstream method types for city-level carbon accounting, which were the pure-geographic production-based method, consumption-based method, and community infrastructure-based method. However, there have been very few digital tools or platforms for city CF calculation. One direction of tools that can be referred to is landscape-oriented tools such as i-Tree Planting Calculator [58] and Pathfinder [59], which allow for a city-scale calculation but are limited to greening elements. Another direction is to apply more general tools that aim for all-industry use, such as Umberto [60] and CarbonStop [61], or employ a consulting service from the corresponding affiliations to assess the carbon-producing process. In general, the development of digital tools in the city scope is at an initial stage.

### 3.2. *Cross-Scope Analysis and Comparison of the Digital Tools*

3.2.1. CF Analysis Strategy

By analysing and comparing the digital tools from the three scopes, the CF calculation methods are mainly three types: the "cradle-to-completed construction" measurement of embodied carbon emissions, the "cradle-to-grave" LCA measurement, and WBLCA for building assets. A total of 3/6 tools in the infrastructure scope and 11/18 tools in the building scope adopt LCA/WBLCA. Because a city or region is dynamic and cannot be defined as before or after construction, only LCA has been considered in previous research. Particularly, several currently dominant digital tools such as One Click LCA (including Infrastructure LCA), Athena series, and Tally [14,16] all employ LCA/WBLCA. There was an argument that in the infrastructure scope, operational carbon emissions only comprised about 3%, whereas the rest comprised all embodied emissions [62]. This might hurdle the LCA adoption in the infrastructure scope. However, researchers pointed out that the "burden shifting" existed if just the embodied emissions were considered in the infrastructure scope [14]. This implied that if we only reduced the embodied emissions, the operational emissions would be increased greatly [14]. Moreover, it was proven that the emissions in the operational phase were much higher than the embodied emissions in the building sector, which induced the same consideration in infrastructure [63]. Therefore, in general, a life cycle CF analysis (i.e., LCA/WBLCA) is still the trend to develop the digital tools.

Nevertheless, the inconsistency of LCA/WBLCA in infrastructure, building, and city scopes can also be a big concern. To be specific, although in this study we defined the CF calculation methods as embodied carbon emissions, LCA, and WBLCA, the actual calculation details (e.g., goals and scopes, system boundaries, functional units, data sources, calculation specifications, etc.) may vary greatly depending on the standards and protocols that the tool follows [17,19]. Even the boundaries of carbon-related terminology could be a blur in the first place. For instance, [14] presented a discussion on the definition of "embodied carbon"; other examples include "embodied carbon vs. capital carbon" and "direct/indirect carbon vs. embodied and operational carbon". The inconsistency has influenced the data standardisation, accuracy of results, and comparisons among different projects, which lead to difficulties for researchers', policymakers', and practitioners' decision making in the long run [14,16,19,52].

Additionally, there is an issue worth noticing, which is the calculation engine transparency of the digital tools. It is difficult to figure out how the analysis is performed specifically (e.g., calculation specifications, applied database, parameters, etc.) using the tools. Especially for developed software tools, although standards or protocols that the tools depend on can be indicated in the product manual, the calculations cannot be investigated or modified. Comparatively, the spreadsheet-based tools work better because the output results can be examined by clicking the cells to check the calculation functions (e.g., the National Highway tool [26]). Several other studies have indicated that the uses of CF calculation tools are mostly project, objective, or scope-based because the databases used by the tools would vary greatly and lead to different results [64,65]. This might cause users to question the reliability of the selected CF calculation tool. Hence, the transparency of the tools should be valued.

3.2.2. Standards and Protocols

The standards and protocols that are covered by the tools in this study are summarised in Table 4. Tracing back to the origins, ISO 14040 and ISO 14044 are the earliest guides that formulate the LCA method, addressing the environmental impacts of manufacturing products in general [33]. Although researchers argued that ISO 14040 and ISO 14044 which were developed 15 years ago might be "old" to some extent [16], they have been the backbones of many digital tools. More recently, standards and protocols for specified scopes have been published to address the more specified carbon issues. For example, PAS 2080 by BSI was published to solve the carbon management issue in the infrastructure

scope; ISO 21900 series was published for building scope CF issues; and PAS 2070 and GPC were designed for city-level carbon issues.

**Table 4.** Summary of CF standards and protocols.

| Standard Institution | Standard Name | Introduction | Launched/ Updated | Reference |
|---|---|---|---|---|
| British Standard Institute (BSI) | PAS 2050 | Specification for the assessment of the life cycle GHG emissions of goods and services | 2011 | [66] |
| | PAS 2060 | Specification for the demonstration of carbon neutrality | 2010 | [67] |
| | PAS 2070 | Specification for the assessment of GHG emissions of a city, direct plus supply chain and consumption-based methodologies | 2014 | [53] |
| | PAS 2080 | Carbon management in infrastructure, a global standard for managing infrastructure carbon, be authorised to meet World Trade Organization requirements | 2016 | [13] |
| International Organization for Standardization (ISO) | ISO 14040 | Environmental management, life cycle assessment, principles and framework | 2006 | [11] |
| | ISO 14044 | Environmental management, life cycle assessment, requirements and guidelines | 2006 | [68] |
| | ISO 14067 | GHG—carbon footprint of products—requirements and guidelines for quantification | 2018 | [69] |
| | ISO 14025 | Environmental labels and declarations, type III environmental declarations. Principles and procedures | 2006 | [70] |
| | ISO 21929 | Sustainability in building construction—sustainability indicators—part 1: framework for the development of indicators and a core set of indicators for buildings | 2011 | [71] |
| | ISO 21930 | Sustainability in buildings and civil engineering works—core rules for environmental product declarations of construction products and services | 2017 | [72] |
| | ISO 21931 | Sustainability in building construction—framework for methods of assessment of the environmental performance of construction works—part 1: buildings | 2019 | [73] |
| | BS EN 15978 | Sustainability of construction works—assessment of the environmental performance of buildings—calculation method | 2012 | [74] |
| | CEN EN 15603 | Energy performance of buildings, overall energy use and definition of energy ratings | 2008 | [75] |
| | CEN EN 15804 | As the EPD standard for the sustainability of construction works and services, describes the technical performance of a construction product, provides data on a set of indicators for each of the different life cycle stages of the product | 2012 | [76] |
| | BS EN 15942 | for business-to-business communication to ensure a common understanding through consistent communication of information for sustainability of construction works | 2021 | [77] |
| Greenhouse Gas (GHG) | GHG Protocol | Establishes comprehensive global standardised frameworks to measure and manage GHG emissions from private and public sector operations, value chains, and mitigation actions | 2015 | [78] |
| IPCC | IPCC Guidelines | IPCC Guidelines for National Greenhouse Gas Inventories | 2006 | [79] |
| WRI | GPC | The Global Protocol for Community-Scale Greenhouse Gas Emission Inventories | 2014 | [54] |

However, iterated methodologies based on the standards and protocols are always coming out. Because, firstly as stated in in Section 3.2.1, the problem of inconsistency is one of the outcomes of having a variety of standards and protocols, and more advanced

methodologies should be developed to amend the gaps. Secondly, [17] stated in the review of city-level carbon accounting that "an inventory of any type of emissions is purely territorial", and similarly, standards and protocols based on the inventories database are territorial to some extent as well. For the newly focused scopes such as infrastructure and city CF, a global standardisation has not been formed yet [19,52]. Therefore, the digital tools should be developed or used consistently by an organisation or project according to local or project characteristics.

### 3.2.3. Rating Systems

In addition to the standards and protocols that the digital tools comply with, multiple building rating systems must also be satisfied by the tools so that the stakeholder can have more comprehensive design support during the project design stage. The common building rating systems are concluded in Table 5.

**Table 5.** Summary of rating systems.

| Name | Affiliation | Content | Ref. |
|---|---|---|---|
| LEED | U.S. Green Building Council | Leadership in Energy and Environmental Design is a green building certification program used worldwide | [80] |
| BREEAM | BRE | Sustainability assessment method for masterplanning projects, infrastructure, and buildings | [80,81] |
| CEEQUAL | BRE | The international evidence-based sustainability assessment, rating, and awards scheme for civil engineering, infrastructure, landscaping, and works in public spaces | [82] |
| Envision | Harvard University | The product of a joint collaboration between the Zofnass Program for Sustainable Infrastructure at the Harvard University Graduate School of Design and the Institute for Sustainable Infrastructure | [83] |
| HQE | GBC Alliance HQE (France), Certivea (Global) | The French certification awarded to building construction and management as well as urban planning projects | [84] |
| Greenstar | Green Building Council Australia | An internationally recognised Australian sustainability rating and certification system for fitouts, buildings, homes, and communities | [85] |
| China Green Building Label (China Three Star) | Research Center of Environment Control and System Optimization | A green building certification program that evaluates projects based on six categories: land, energy, water, resource/material efficiency, indoor environmental quality, and operational management | [86] |

### 3.2.4. Development Level of the Digital Tools in the Three Scopes

From the review results of the three scopes, the digitalisation of CF calculation is more developed in the building sector than the other two. Several mature digital tools have been integrated with BIM software and can be simulated within a digital environment, which would be helpful for the design decision-making collaboration with all stakeholders. In the infrastructure sector, the digitalisation is at a relatively pilot stage, where the tools are turned to be the automated version of calculation specifications to some extent. In the city CF scope, the digitalisation development is more at an infant stage, where the research on theoretical methodology is more focused.

Besides the importance of adaptation to BIM and digital environment availability, other functional features were noticed. For example, the results reporting forms vary from simple CF breakdown numbers to detailed assessment results with automated visualised diagrams. The user interface design of the digital tool is another intuitive feature that indicates the maturity of the tool relatively, but the assessment of this feature can be personal and subjective

to users that the authors excluded it from the summarised table. These features not only affect the design decision-making support but also affect the overall user experience, which is a significant factor in deciding the success or failure of a digital tool [87].

## 4. Discussions

According to the analysis and comparison of the digital tools in the infrastructure, building, and city scopes, respectively, from Section 3, there are several trends that can be concluded to drive the development of digitalisation in carbon management.

### 4.1. The Trend of CF Digital Tools Development Scope

As it can be observed from the reviewed tools of the three scopes, the number of the tools and the level of maturity in the building sector exceed those in the other two scopes. Therefore, a large number of development opportunities exist in the infrastructure and city scopes. Moreover, because of the nature of infrastructure's system complexity, the tools can more deliberately target specific infrastructure assets, such as the railway CF tool, airport CF tool, bridge CF tool, transit system CF tool, etc. For city CF calculation, the tools need to be comprehensive including all necessary urban elements, which may be the current research gap before the tools are well-developed. Although the competition might be fierce in the building scope, there might also be opportunities for CF tools accurately targeting different building types such as residential, commercial, office, or educational buildings.

### 4.2. The Trend of Whole Life Cycle Carbon Analysis

The tendency to utilise a future digital tool to manage the life cycle CF is inevitable. For infrastructure scope, it is crucial for the tool to employ the LCA method to calculate the CF throughout the life cycle of the infrastructure assets, which includes the CF within the "cradle-to-grave" boundary. This could enable a more comprehensive and accurate result of the carbon emissions compared to calculating the embodied carbon emissions only. For the building scope, the transformation between LCA and WBLCA needs to be accelerated to improve the carbon calculation accuracy. Additionally, no matter the scope, the inconsistency issue of carbon assessment should not be overlooked. Efforts should be taken both from the industrial standards and protocols aspect and academic methodology aspect. In order to better assist stakeholders in the decision-making process, the digital tools can try to align with a wider range of rating systems.

### 4.3. The Trend of Digitalisation towards a Smarter and Intelligent City

Although the development of cutting-edge technologies such as the Internet of Things (IoT), big data, artificial intelligence (AI), etc. has soared over the years, the CF assessment for existing infrastructure, buildings, and cities has lagged behind. However, the improvement space is large enough for practitioners and researchers to conceive and conduct studies from the following aspects.

Functionally, the adaptation and integration with BIM are necessary for digital carbon tools. The trend can be observed in several current market-dominant tools such as One Click LCA, Tally, and eToolLCD. At the same time, the necessity has also been confirmed academically, especially in infrastructure and building scopes [16,19]. To be more specific, first, the assistance of BIM provides a digital environment that clarifies the complexity of assets' elements and system boundaries [16]. For example, one of the significant features of One Click LCA is that users can import design information models such as Revit files, IFC, and energy model files to conduct CF analysis directly. Second, integrating BIM with CF analysis improves the automation level using the material take-off from BIM and the results can be updated automatically while the design model changes. This was the impetus for the development of Tally [42]. Third, in addition to allowing BIM data input for CF analysis, the trend is also towards developing a digital carbon management tool based on BIM, which regards BIM as the data exchange environment [16]. Academic pilot studies have been conducted for a BIM-based carbon LCA tool for life cycle cost and BIM-based real-time

platform for green building rating [88,89]. This tendency has gradually merged with the development of digital twin for the whole life cycle asset management, which might extend the CF analysis from a single "function" into a critical system of asset management. In general, the integration of BIM and CF is beneficial and inevitable.

Technologically, several improvements are suggested to bridge the technology application gap. First, as the requirements of carbon management gradually become intricate, advanced software development techniques should be adopted. Although the release form such as spreadsheets and standalone software can satisfy users' needs in the first place and can be developed quickly, the evolving requirement of multi-stakeholder collaboration and accessibility to cloud services (e.g., cloud computing) and web-based applications turns to be the trend of digital tools. Thus, development toolkits such as JavaScript and CSS are recommended. Moreover, better user experiences and user interface designs would assist the maturity of the digital tools. Second, given that the carbon issue gradually becomes not only an environmental and sustainable problem but also more of an economic and even political issue as it regards carbon trading and policymaking [33], emerging techniques such as digital twin and IoT can enable real-time monitoring of the physical built environment and thus help the decision-making process. AI, machine learning, and blockchain based on a large amount of CF analysis data are very possible to solve environmental issues [90,91]. The technological implementation of carbon issues would drive the building, infrastructure, and city to smarter and more intelligent ones.

Furthermore, the information management aspect should also be covered by the future development of digital carbon tools. The reasons come from both the industrial practitioners and the drivers of technology. On the one hand, the rooted nature of the AEC/FM industry, such as organisational complexity, lack of innovation, and data paucity in the infrastructure sector, can also hinder the collection and processing of the data for CF analysis. Furthermore, due to the regional feature of standards and protocols, the comparisons and improvements using the analysed results would be more difficult [19]. These lead to the isolation of information and feedback, which strengthen the difficulty to foster and iterate a comprehensive and efficient digital carbon management tool. On the other hand, the application of technology can improve the novelty of the digital carbon tools but can also require more efficient information management to provide effective data to use the technology. Therefore, proper management such as unified data sources, a shared database of the carbon-related supply chain, a common data environment for data updating and sharing, data security, etc. needs to be a concern while developing the digital carbon tool.

Lastly, the organisational and operational aspect should not be overlooked during the digital transformation. During this study, it was observed that the tools that offer proper product guidance and training materials are more accessible both to target users and researchers. This could be learned for future digital carbon tool developers. Moreover, training and activities for the public can drive the operation of a carbon community, such as the current BIM community for researchers and practitioners, which is bi-directionally beneficial between the industry and individual.

## 5. Conclusions

This study addressed the crucial environmental issue of CF reduction in the AEC/FM sector from the digitalisation perspective. To reveal the issue, this paper proposed a four-step methodology to review the academic and commercial digital tools for CF calculation targeting infrastructure, building, and city scopes. The methodology included (1) identifying review scopes and data sources, (2) confirming searching keywords, (3) screening and preliminary documentation, and (4) summarising 14 shared attributes in three types (i.e., basic information, CF analytical information, and digitalisation information). To present the results, all the reviewed tools were summarised in the spreadsheet format shown in Tables 2 and 3. Analysis of the tools was conducted in terms of the infrastructure, building, and city scopes. Moreover, the comparison and generalisation of the tools crossing the

three scopes are presented. Based on the results and analysis, the overall conclusions were as follows:

(1) There are mature and effective digital tools targeting CF analysis in infrastructure, building, and city scopes covering most of the analysis in the AEC/FM sector in general. However, the development level is uneven. Comparatively, the tool choices for building scope scenarios are greater than for infrastructure and city scopes.

(2) The current CF analysis methods are swaying between embodied carbon emissions calculation and whole-life carbon calculation (i.e., LCA and WBLCA). As the carbon issue becomes more and more severe confronting the climate change challenge, a more accurate whole-life carbon assessment is the mainstream. Moreover, issues such as calculation inconsistency still exist to be addressed in the current assessment methodology.

(3) The current digital tools have realised the automation of CF calculation, but the development level is far from "smart" or "intelligent". The tools cannot be easily adapted and adopted to other digital approaches such as BIM or digital twins populating in the AEC/FM sector.

Future development trends are proposed for researchers and practitioners to discuss:

(4) Advanced tools and approaches in digital forms targeting multiple stakeholders in the infrastructure and city scopes are welcomed to fill the current gap.

(5) Accurate and consistent CF assessment methodologies and globalised standards and protocols should be developed in each scope, focusing on the whole-life cycle carbon assessment. Future digital tools should be implemented in actual scenarios to provide more empirical experiences and feedback.

(6) The digitalisation of CF assessment in the AEC/FM sector should be developed towards a smarter and more intelligent city goal. Efforts can be taken from aspects of functionality, technology, information management, and organisation and operation.

Overall, the urgent environmental issue is the foremost concern for all the nations in all disciplines. As the AEC/FM sector is responsible for a large number of carbon emissions, we should take efforts to tackle this emergency. The step towards digitalisation would be a great improvement.

**Author Contributions:** Conceptualisation, Q.L. and L.C.; methodology, J.Y. and Q.L.; validation, Q.L. and J.H.; writing—original draft preparation, J.Y.; writing—review and editing, Q.L., J.T., L.C. and J.H.; supervision, Q.L. and J.T.; project administration, T.B.; funding acquisition, Q.L. and J.T. All authors have read and agreed to the published version of the manuscript.

**Funding:** This research is supported by the UCL-PKU Strategic Partner Funding (Project code: 556103.179601) Scheme 2021/22, and ICE Research and Development Enabling Fund (Project code: 572340.184864).

**Institutional Review Board Statement:** Not applicable.

**Informed Consent Statement:** Not applicable.

**Data Availability Statement:** Not applicable.

**Acknowledgments:** This work was adopted from the conference paper "SeeCarbon: a review of digital approaches for revealing and reducing infrastructure, building and City's carbon footprint" that will be presented on IFAC AMEST 2022, 26 July–29 July 2022.

**Conflicts of Interest:** The authors declare no conflict of interest.

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
