# Peer review of "Digital Tools for Revealing and Reducing Carbon Footprint in Infrastructure, Building, and City Scopes"

_buildings, doi:10.3390/buildings12081097_

Round 1

Reviewer 1 Report

English language review needed. For example, in the abstract “… digital technology brings the opportunity to migrate environmental issues”, I believe “migrate” should be mitigate.

Table 1 needs additional horizontal lines to distinguish which text goes where. It is difficult to determine which attribute matches which content, as some of the lines blur together.

One subject within analytical approach for the programs not discussed enough is how transparent the programs are in their calculations. Some of these software, such as Athena and Tally, use proprietary calculation engines where it is difficult to change or see which EPDs are being used and how calculations regarding end of life, beyond life, etc. are being performed. This is important because several other studies have shown how much different LCA results can be on identical Bill of Materials depending on the software package. This issue is briefly addressed in the conclusions.

In section 4.2 I am confused by the sentence “it is crucial for the tool to employ LCA method instead of “cradle-to-complete construction” method…”. LCA is life-cycle analysis, which is an analysis type, and “cradle-to-complete construction” is a boundary system within LCA. Some LCA’s are done production stage only “cradle-to-gate”, others are done over large boundary systems, but they are all LCA’s. It seems like you are trying to imply that full life-cycle boundaries should be required “cradle-to-grave”, but the section doesn’t read clearly that way. Please revise language to make your argument clearer.

Reviewer 2 Report

Dear Authors,

please address the followings:

lines 50-52. the sentences is maybe missing a verb and difficult to grasp. Please, revise it. 

lines 64-66. the sentences is missing a verb. Please, revise it. 

tables 2 and 3. it would be beneficial to state clearly which LCA stages (A1-A5, etc.) are included in each tool.
